genomics

taxonomic bias, model organisms, massively parallel sequencing, ethnography of scientists, species conservation status

**Authors for correspondence:**
Margarita Hernandez
e-mail: mzh235@psu.edu
George H. Perry
e-mail: ghp3@psu.edu

### PUBLISHING

# Factors influencing taxonomic unevenness in scientific research: a mixed-methods case study of non-human primate genomic sequence data generation

Margarita Hernandez[1], Mary K. Shenk[1]
and George H. Perry[1,2,3]

[1]Department of Anthropology, [2]Department of Biology, and [3]Huck Institutes of the Life Sciences, Pennsylvania State University, University Park, PA 16802, USA

MH, 0000-0001-6522-6455; MKS, 0000-0003-2002-1469; GHP, 0000-0003-4527-3806

Scholars have noted major disparities in the extent of scientific research conducted among taxonomic groups. Such trends may cascade if future scientists gravitate towards study species with more data and resources already available. As new technologies emerge, do research studies employing these technologies continue these disparities? Here, using non-human primates as a case study, we identified disparities in massively parallel genomic sequencing data and conducted interviews with scientists who produced these data to learn their motivations when selecting study species. We tested whether variables including publication history and conservation status were significantly correlated with publicly available sequence data in the NCBI Sequence Read Archive (SRA). Of the 179.6 terabases (Tb) of sequence data in SRA for 519 non-human primate species, 135 Tb (approx. 75%) were from only five species: rhesus macaques, olive baboons, green monkeys, chimpanzees and crab-eating macaques. The strongest predictors of the amount of genomic data were the total number of non-medical publications (linear regression; $r^2 = 0.37$; $p = 6.15 \times 10^{-12}$) and number of medical publications ($r^2 = 0.27$; $p = 9.27 \times 10^{-9}$). In a generalized linear model, the number of non-medical publications ($p = 0.00064$) and closer phylogenetic distance to humans ($p = 0.024$) were the most predictive of the amount of genomic sequence data. We interviewed 33 authors of genomic data-producing publications and analysed their responses using grounded theory. Consistent with our quantitative results,

authors mentioned their choice of species was motivated by sample accessibility, prior published work and relevance to human medicine. Our mixed-methods approach helped identify and contextualize some of the driving factors behind species-uneven patterns of scientific research, which can now be considered by funding agencies, scientific societies and research teams aiming to align their broader goals with future data generation efforts.

## 1. Introduction

Scholars have long observed taxonomic unevenness in terms of focal species included in published research studies. On a broader taxonomic level, birds and mammals are over-represented in the scientific literature, while fish, amphibians and invertebrates are included at a relative deficit to their actual abundance in nature [1–3]. Conservationists specifically have observed the tendency for species to be selected based on their 'charisma' or appeal (for reasons society may ascribe to certain species based on their 'beauty, valor or singularity') to scientists and/or the general public [4–6]. Additionally, species that are characterized as 'models' for various processes or fields—for example *Arabidopsis thaliana* in the botanical sciences or rhesus macaques (*Macaca mulatta*) in biomedicine—may continue to be disproportionately studied due to the benefit from the continuous accumulation of knowledge and research tools specific to that organism [7].

These patterns of taxonomic unevenness in scientific research matter. Specifically, future scientists will be primed to more readily and powerfully answer novel questions with species having extensive histories of prior study relative to more understudied taxa. This cascade is especially strong when the data produced in earlier studies have been made freely available to other researchers; in addition to reproducibility-related benefits, public data sharing allows for important, downstream research questions to be developed and answered using data originally generated for other research purposes.

For our study, we sought to assess whether the longstanding taxonomic unevenness in scientific research publication is similarly observed in patterns of emerging technology use. If so, then what factors are influencing or even driving this phenomenon? Conveniently, given the still-growing use of the technology that is the focus of our study, we can investigate these potential patterns of unevenness in real time and incorporate insights from interviews with the very scientists generating these data.

Specifically, we focused on the use of massively parallel genomic sequencing methods. The development and continued technological innovation of these tools have helped scientists answer expanding sets of questions in species biology, evolutionary history, behavioural ecology and population dynamics [8–11]. The genetics and genomics community as a whole has been a leader in the data sharing movement, with standardly used online repositories including the National Center for Biotechnology Information's Sequence Read Archive (SRA), the Gene Expression Omnibus and GenBank [12].

Our study aims to investigate patterns of taxonomic unevenness within publicly available genomic sequence data archives, using non-human primates as a case study. Non-human primates are among the world's most endangered taxonomic groups, with 60% of all non-human primates at risk of extinction [13,14]. Non-human primates serve important ecological, cultural and medical purposes [13]. Their extinction would threaten the ecosystems they inhabit and our opportunities to understand human biology. Given their close phylogenetic relationship with humans, non-human primate taxa have been regularly studied to help understand the progression of many human diseases [15,16], including HIV [17] and Alzheimer's [18].

Our goal was to identify variables associated with patterns of species-unevenness in genomic sequence data across all 519 non-human primate species. Are individual predictors (or combinations thereof) such as non-medical publication history, medical publication history, geographical range, frequency in captivity, International Union for the Conservation of Nature (IUCN) Red List conservation status, activity pattern and phylogenetic distance to humans significantly associated with patterns of genomic data availability? Additionally, we incorporated a qualitative component in which we interviewed first and/or corresponding authors on papers that generated non-human primate genomic sequence data to record their motivations and the factors that they explicitly considered when selecting species to study. This mixed-methods approach let us identify quantitative patterns in the existing distribution of published genomic sequence data while simultaneously investigating the contexts in which these data were generated.

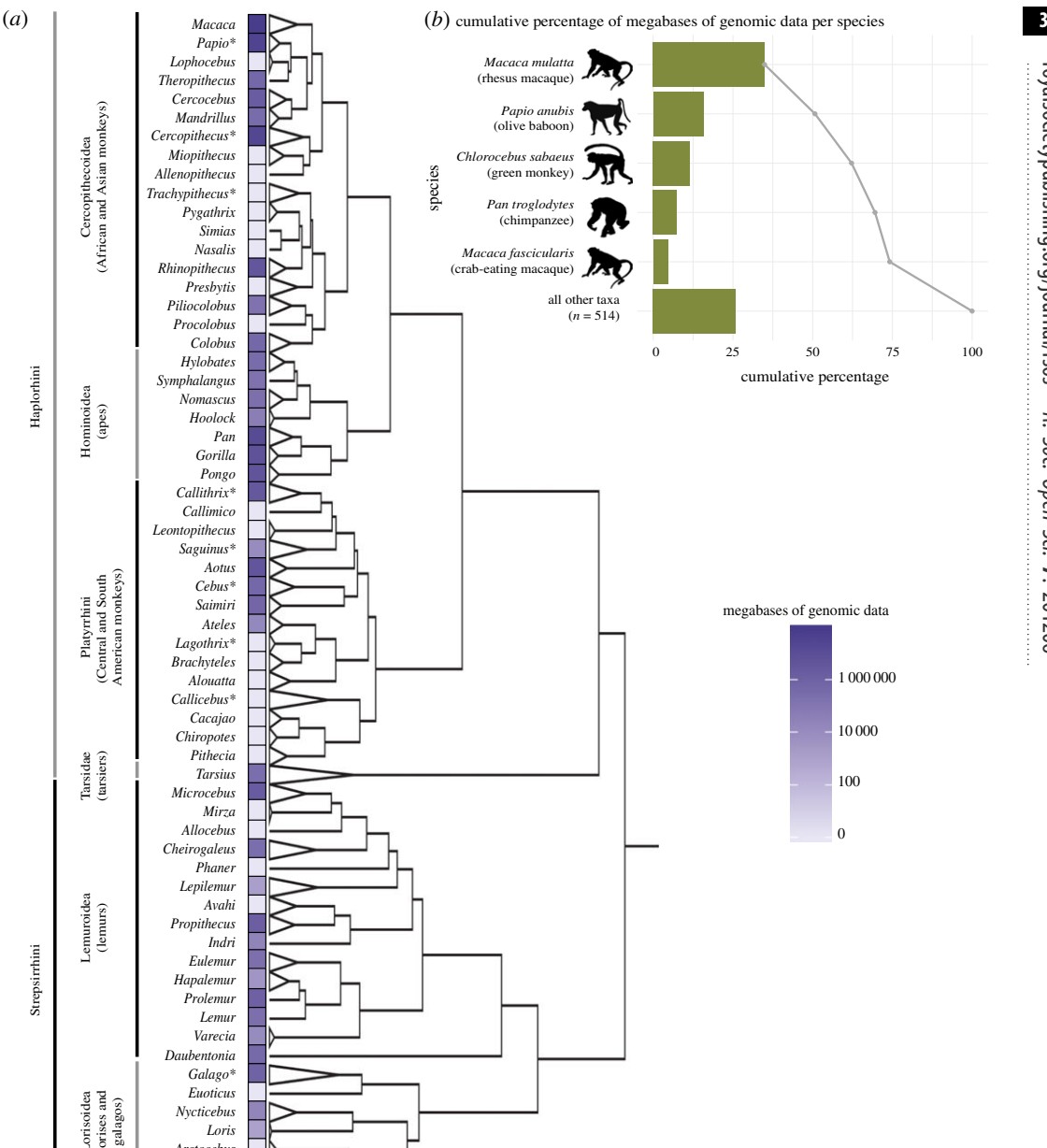

**Figure 1.** Megabases of genomic data by genus across the order Primates and for the five species with the most genomic data. (*a*) Phylogeny of the order Primates with dark purple indicating more genomic data per genus and white indicating little to no genomic data. Paraphyletic genera are denoted with an asterisk. A complete list of genera is provided in electronic supplementary material, table S1. Phylogeny adapted from Dos Reis *et al*. [19]. (*b*) The five species with the most genomic data and the cumulative percentage of the total amount of non-human human primate sequence data represented by these taxa. Credit to T. Michael Keesey and Tony Hisgett for the chimpanzee image, under license https://creativecommons.org/licenses/by/3.0/.

## 2. Results

We downloaded metadata for a total of 179.6 terabases (Tb) of non-human primate genomic sequence data available in the NCBI Sequence Read Archive (SRA) database as of 16 August 2018. The order Primates comprises a total of 520 species (including humans). We found that 416 of the 519 (80.2%) non-human species did not have *any* genomic sequence data deposited in SRA at the time of our analysis. Of the 103 (19.8%) species that are represented, the majority of the sequence data (133.2 Tb; 74.2%) come from only five different species (figure 1): rhesus macaques (*M. mulatta*), olive baboons (*Papio anubis*), green monkeys (*Chlorocebus sabaeus*), chimpanzees (*Pan troglodytes*) and crab-eating macaques (*Macaca fascicularis*).

For each non-human primate species, we also recorded the following information based on a combination of our own hypotheses and variables considered in previous species disparity studies in other taxonomic groups [20–24]: current conservation status (least concern, near threatened, vulnerable, endangered and critically endangered) and geographical species range (km$^2$) from IUCN, the number of both medical and non-medical scholarly publications featuring each species from the Web of Science database, the estimated evolutionary distance to humans (millions of years ago for most recent common ancestor) from a recent phylogenetic analysis [19], the number of individuals currently housed in more than 1000 worldwide zoos and other conservation facilities who are Species 360 members, and activity pattern (nocturnal, diurnal, cathemeral) [25] (electronic supplementary material, dataset 1). These variables were compared to the amount of genomic data (Mb) available in the SRA database for each species, both on an individual variable basis (e.g. linear regressions) and collectively (e.g. logistic regression). We also performed linear regressions among all combinations of predictor variables (electronic supplementary material, figure S3).

## 2.1. Variables associated with the presence or absence of genomic data

First, given the large proportion of non-human primate species without any available genomic data ($n = 416$), we tested which variables were significantly associated with the presence versus absence of genomic data (electronic supplementary material, figure S1). We found that species with genomic data, as a group, have significantly more non-medical publications (average $763.26 \pm 2915.19$ s.d.) than those without genomic data ($28.82 \pm 74.42$; Mann–Whitney $U$-test; $p < 2.2 \times 10^{-16}$) (electronic supplementary material, figure S3A). We also observed a significant difference in the number of medical publications between species with genomic data ($16.17 \pm 84.56$) compared to those without genomic data ($0.11 \pm 1.08$; $p < 2.2 \times 10^{-16}$) (electronic supplementary material, figure S3B). Species with genomic data tended to be more closely related to humans (millions of years since last shared common ancestor with human; $45.03 \pm 20.97$) than those without genomic data ($48.80 \pm 17.17$; $p = 0.0125$) (electronic supplementary material, figure S3C). Finally species with genomic data available also had larger geographical ranges ($905\,615 \pm 1\,664\,226$ km$^2$) and more individuals in captivity ($242 \pm 502$ individuals) than species without genomic data ($385\,345 \pm 815\,673$ km$^2$; $p = 0.00429$; $28 \pm 98$ individuals; $p < 2.2 \times 10^{-16}$) (electronic supplementary material, figure S3D and E). Presence/absence of genomic data were also significantly associated with Red List status ($\chi^2$-test; $p = 0.0024$) but not with activity pattern ($p = 0.1506$). We also performed a logistic regression and determined that a greater number of non-medical publications ($p = 2.57 \times 10^{-7}$), a greater number of individuals in captivity ($p = 0.0411$), and species categorized as endangered relative to critically endangered within IUCN Red List status ($p = 0.0229$) were significantly predictive of the presence of genomic data in the context of all other variables.

## 2.2. Variables associated with the amount of genomic data per species

In addition to the presence/absence of genomic data, we tested whether the amount of genomic data (megabases in the NCBI SRA database) per species is significantly associated with our variables of interest (figure 2). For the entire dataset (including species with no genomic data), the total number of non-medical publications explained 33% of the variation ($r^2 = 0.33$) within the genomic sequence data ($p < 2 \times 10^{-16}$) (figure 2a). Number of medically focused publications and frequency in captivity explained 27% and 22% of the variation within genomic sequence data ($p < 2 \times 10^{-16}$ and $p < 2 \times 10^{-16}$, respectively) (figure 2b,e). While phylogenetic relatedness to humans and geographical range were statistically significantly associated with genomic data, they had limited explanatory power ($p = 5 \times 10^{-6}$ and $r^2 = 0.038$, and $p = 0.00032$ and $r^2 = 0.028$, respectively) (figure 2c,d). IUCN Red List status was neither significant nor explanatory (IUCN Red List status treated as an ordinal variable; $p = 0.926$, $r^2 = -0.0021$) (figure 2f). Using ANOVA, there were no statistically significant differences across IUCN Red List or activity pattern categories in the amount of genomic sequence data available ($p = 0.244$ and $p = 0.49$, respectively).

We also performed a generalized linear model under a Gaussian distribution to identify variables that best predicted the amount of genomic sequence data per species while accounting for interdependence among these factors (see Material and methods). Based on this model, non-medical research publications ($p = 7.62 \times 10^{-9}$), number of medical publications ($p = 3.74 \times 10^{-9}$), number of individuals in captivity ($p = 0.022$) and species categorized as endangered relative to critically endangered within IUCN Red List status ($p = 0.026$) were all significant predictors of the amount of genomic sequence data ($p <$

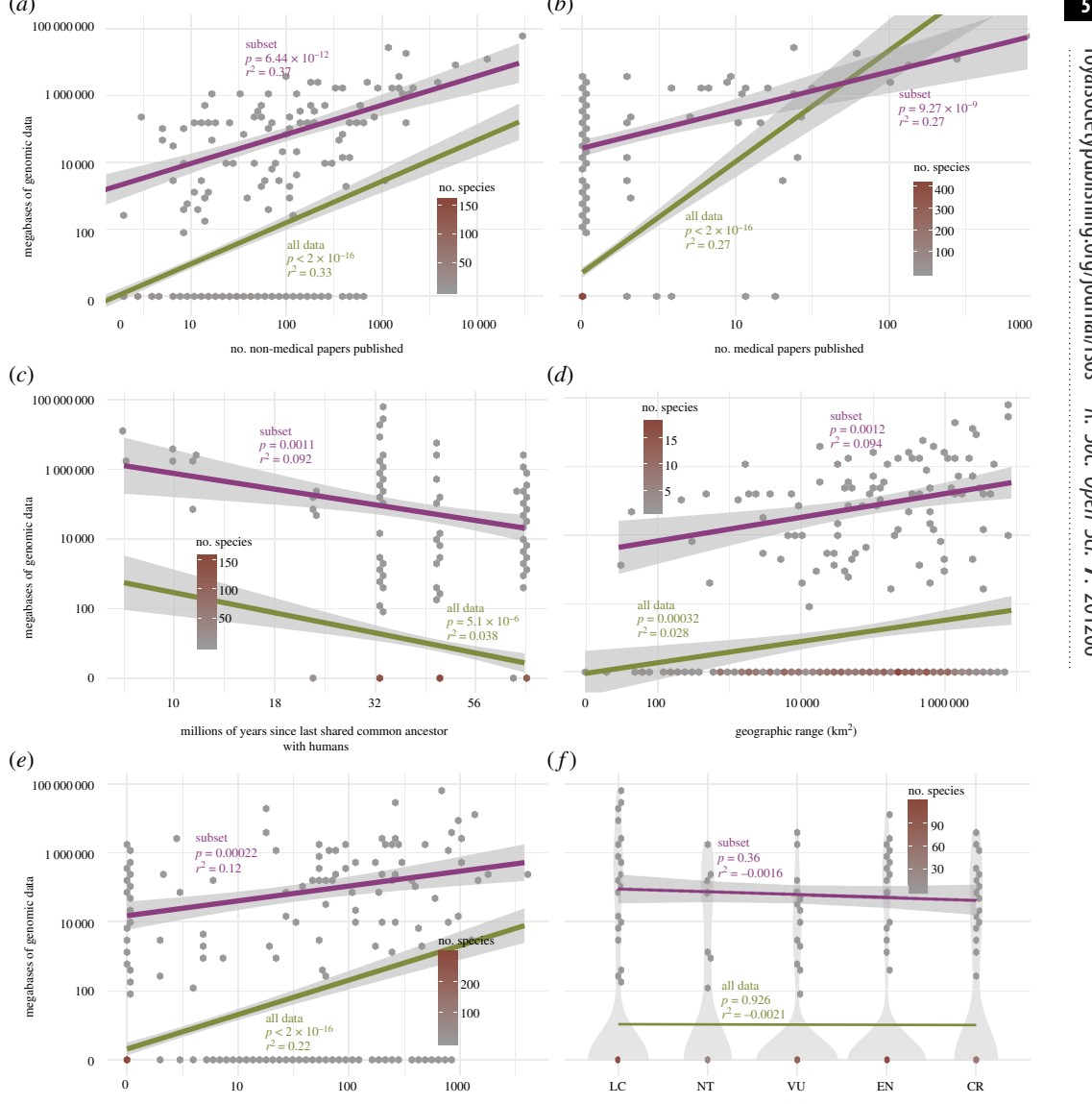

**Figure 2.** Linear regressions for the entire dataset and subset of species with genomic data. Linear regressions for six variables used within the study, non-medical papers published (*a*), importance in medical research (*b*), relatedness to humans (*c*), geographical range (*d*), frequency in captivity (*e*) and IUCN Red List status (*f*). For each, the purple line represents the linear regression for the subset of species with genomic data available, while the green line represents the linear regression for all species within the dataset.

$2.2 \times 10^{-16}$; $r^2 = 0.40$). To help assess our model, we performed the leave-one-out cross validation (LOOCV) analysis to compare predicted megabases of genomic data from our generalized linear model to the observed results (electronic supplementary material, figure S4A). Predicted and observed values were significantly correlated (linear regression; $r^2 = 0.37$; $p < 2.2 \times 10^{-16}$). We obtained similar results when restricting the analysis to the subset of species with genomic data ($r^2 = 0.32$; $p = 5.93 \times 10^{-10}$; electronic supplementary material, figure S4B).

Since the normality assumption for linear regressions was violated in our analyses of the full dataset, we also repeated these analyses on the subset of the dataset with species having at least some (i.e. non-zero) genomic data (figure 2). In this analysis, total number of non-medical publications explained approximately 37% of the variation in genomic sequence data ($p = 6.44 \times 10^{-12}$) (figure 2*a*). Number of medically focused publications was also significant and explained 27% of the variation within genomic sequence data ($p = 9.27 \times 10^{-9}$) (figure 2*b*). Relatedness to humans ($p = 0.00106$), geographical range ($p = 0.0012$) and frequency in captivity ($p = 0.00022$) were all statistically significant but had limited explanatory power ($r^2 = 0.092$, $r^2 = 0.094$ and $r^2 = 0.12$, respectively) (figure 2*c–e*). IUCN Red List status was neither significant nor explanatory ($p = 0.361$, $r^2 = -0.0016$) (figure 2*f*). Our generalized

linear model with the subset of species with genomic data present revealed that number of non-medical publications ($p = 0.00064$) and relatedness to humans ($p = 0.024$) were significant predictors of the amount of genomic sequence data ($p = 6.16 \times 10^{-8}$; $r^2 = 0.40$).

## 2.3. Author motivations in selecting species for study

We randomly selected 300 unique SRA study numbers with the goal to contact the corresponding authors on papers for which these data were originally generated. We invited 216 authors (as some deposits did not have an associated publication and some individuals were corresponding authors on multiple papers) to participate in a semi-structured interview. In total, and after obtaining informed consent, we conducted 33 semi-structured interviews with first and/or corresponding authors on 33 publications that generated non-human primate genomic sequence data represented in our database. The 15.3% response rate is within the typical range for email/Internet surveys [26]. The list of interview questions is presented in electronic supplementary material, table S2. We analysed major themes arising from the interviews using a grounded theory approach [27]. Twelve themes emerged from this analysis, grouped into four categories: opportunistic research, interest in species, human implications and methods development (figure 3).

### 2.3.1. Opportunistic research

The authors frequently mentioned selecting species with sampling and analytical feasibility in mind. Four themes were categorized under opportunistic research: ACCESS (present in $n = 26$ of 33 total interviews), HISTORY ($n = 23$), CAPTIVE ($n = 10$) and REFER ($n = 11$). Having access to high-quality existing samples, the availability of easily acquired cell lines and/or access to captive individuals were repeatedly mentioned as being important. Many authors also mentioned that species with extensive histories of prior work, in turn, helped them to more feasibly conduct analyses (e.g. due to the presence of a high-quality, annotated reference genome), or better contextualize their own results:

> [The taxa] had data, behavioral data…hormonal data, they had super early genetic markers. It was a system that had been studied from multiple different angles, multiple Ph.D. students and the like worked on it, so it was good because we had some hypotheses for what we'd find different between the two taxa.

### 2.3.2. Human implications

The second major category that emerged from our interviews was human implications. Authors frequently mentioned that their research questions were directly relevant to questions regarding human biology, and more specifically human diseases. The most repeated theme in this category was a specific mention of species being used to understand human phenomena. This category contained three themes: HUMAN ($n = 21$), RELATED ($n = 12$) and MODEL ($n = 10$). One author commented:

> One of the primary goals of biomedical research is to develop animal models which will allow us to better understand the causes and potential treatments for human diseases. And so if you understand that genetics is important for human disease, and you want to model diseases in a non-human primate, then clearly understanding genetics and genetic differences among rhesus macaques is going to be important in a couple different ways.

### 2.3.3. Interest in species

Another emergent category was the author interest in the species. This category contained four themes: COLLAB ($n = 15$), CONSER ($n = 12$), PERMITS ($n = 13$) and OPTION ($n = 8$). Multiple authors mentioned conservation implications as either a priority and/or a by-product of their research questions. Some researchers specifically selected certain species because of their IUCN Red List status (e.g. critically endangered, endangered, etc.) and still other authors selected species primarily for different reasons but with conservation implications also in mind. Many of the same (and other) authors also frequently mentioned difficulties in securing permits and ensuring that local governmental regulations were properly followed, especially when studying protected species. Collaborations with other research groups and international scientists was, in some cases, critical for the continuation and completion of the research work, as one author commented:

> I may not have necessarily continued doing this if I wasn't given the opportunity through collaborators.

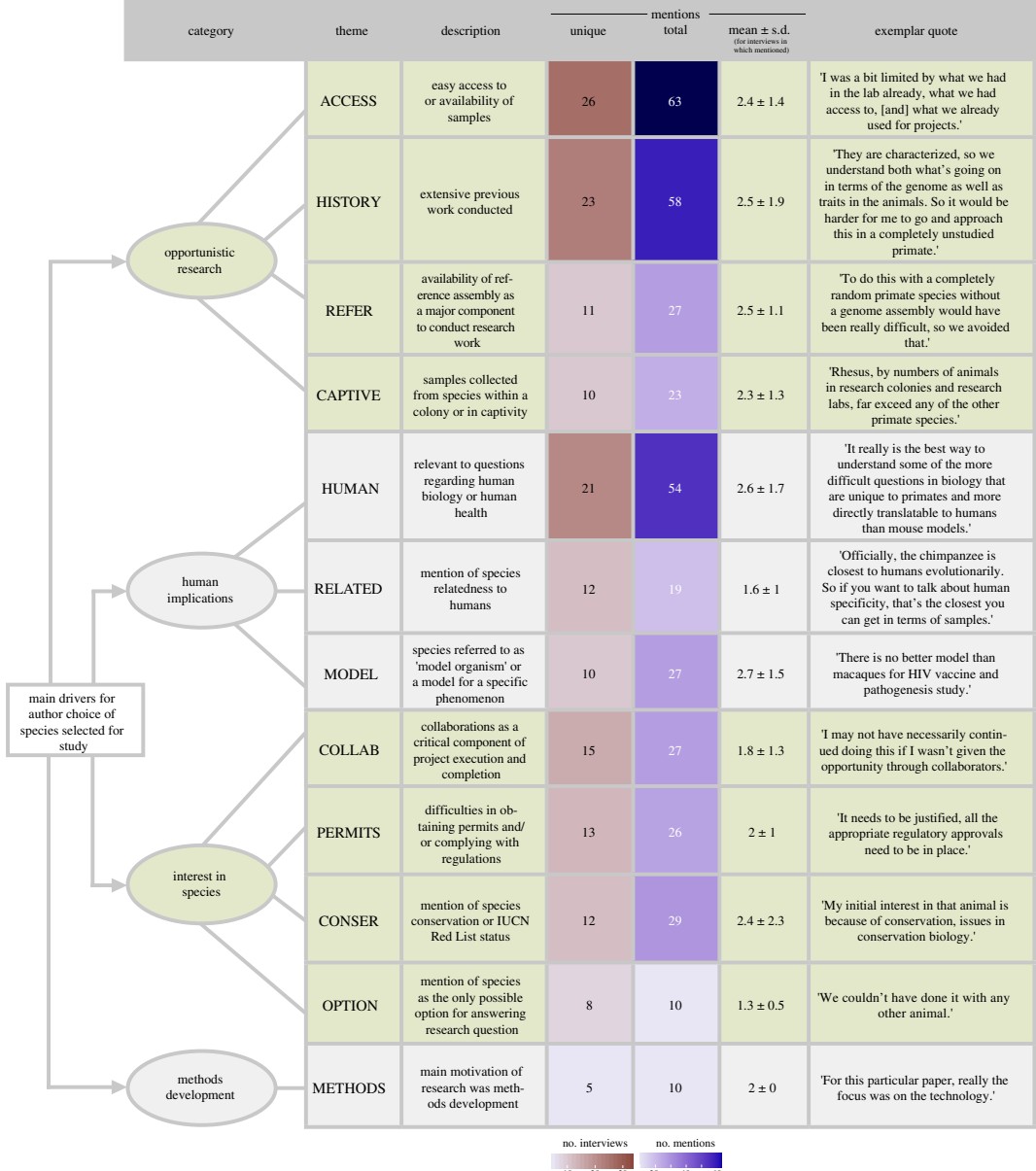

**Figure 3.** Main drivers for author choice of species selected for study. Each theme derived from our grounded theory analysis listed with its description, the number of unique interviews the theme appeared in (max 33), the total number of times each theme appeared, the mean and standard deviation across interviews where the theme was mentioned at least once, and an exemplar quote. Themes are organized by comprehensive categories that inform author choice when selecting non-human primates for research studies. The heatmaps depict the number of interviews each theme was present in and the number of total mentions for each theme.

### 2.3.4. Methods development

The final category only consists of one theme, METHODS ($n = 5$). These authors mentioned that, in some cases, their research study was driven primarily by an interest in developing a new technology, bioinformatics pipeline and/or wet laboratory method. They then used non-human primate samples that were readily available in order to most efficiently develop, evaluate and report on their method.

## 3. Discussion

Our findings contribute to the body of work supporting the idea that certain taxonomic groups, in our case individual non-human primate species, are studied more extensively than others. Specifically, certain non-human primates appear to have been selected for massively parallel genomic sequencing

studies primarily because their biological samples were available and accessible, they had extensive histories of prior published work, and they were relevant to questions pertaining to human biology, especially when investigating the human diseases. As our closest living relatives, non-human primates are researched extensively to help us better understand fundamental questions regarding our evolutionary history and the diseases that plague us [17,28]. We note that there were many more non-medical than medical publications in our quantitative analysis. Thus, conclusions regarding the relative importance of these two variables should be considered tentative, given power differences. Future studies analysing different sub-types of genomic data (e.g. shotgun genome sequencing, reduced representation sequencing, RNA-sequencing) may provide further insights into the potential drivers of taxonomic unevenness (e.g. [29]).

Our qualitative results aligned closely with those from our quantitative analysis. The qualitative data in particular clearly illustrate how the relative ease of studying certain species already widely used as biomedical models for human disease may further perpetuate future data generation disparities. As one author described:

> If we didn't have macaques, we really wouldn't have a good model for HIV/SIV, and that would be a huge problem. It's not that it's a bad thing that we focused on rhesus monkeys or cynomolgus macaques, but it has consequences just because it might be that stump-tailed macaques are a great model for Parkinson's disease, but we don't know that because we lost all the stump-tailed macaques. It might be that a particular form of spider monkey is a fantastic model for high blood pressure, but we don't know that because there aren't spider monkeys in research colonies that people can study. It's frustrating to me that there are probably outstanding models [of] human disease that we will never discover because we don't have access to those animals. Now that's unfortunate, but you can understand why the NIH can't pay for colonies of 1000 of every different species of primates just because maybe 20 years from now, somebody is going to have a need for those animals. Decisions are driven by resources, by how much you can spend, and you put your resources where you think they will do the most good today. But that sometimes has long term consequences.

We are uncertain of and unqualified to help define the best approach for developing and funding future research on non-human primate models for human disease. It is possible that the current system could be the most efficient and broadly effective. Still, it is important to recognize the manner by which this system further exacerbates patterns of taxonomic unevenness in research—including via new rounds of studies that are not themselves necessarily biomedically motivated—due to enhanced sample accessibility and opportunities to more rapidly advance new research given existing backbones of knowledge on which to build. This phenomenon could in turn constrain opportunities for research on non-model organisms for evolutionary biology, behavioural ecology or conservation purposes. The same consideration probably applies to other taxonomic groups other than primates. That is, even in the absence of model organisms, widely apparent biases for particular field sites [30], geographical regions [31], habitat types [32], species 'charisma' [5] and societal preferences [33] probably impact taxonomic choices in successive research planning processes.

Insights from the qualitative component of our study into the processes that shape scientists' decision-making may aid funding agencies, scientific societies and research consortia whose goals are not fully aligned with the current scientific data generation landscape. Specifically, our study demonstrates that scientific research is goal-oriented and that study organism selection is understandably based to a large degree on feasibility as well as the extent of previous published work and resources. Thus, research-oriented institutions may benefit from taking steps to increase access to biological samples and to develop and disseminate initial genomic-scale data and resources for targeted taxonomic groups. Other researchers would then be more likely to select these species for their own studies (even those funded by other agencies) and create new knowledge and further resources to the positive-reinforcement benefit of all.

# 4. Material and methods

## 4.1. Quantitative data and analyses

### 4.1.1. Non-human primate species list

We generated a list of all non-human primate species using the IUCN Red List and supplemented using *All the World's Primates* by Rowe and Myers [25]. Thirteen species were found only within the IUCN Red List, 82 species were found only within Rowe and Myers, 396 species were found in both sources and 29 species were found in both sources but under synonymous species names. Using these two sources, we

arrived at a list of 519 species. For the purposes of this study, we collapsed any subspecies under a single species name. All data recorded for each species can be found in electronic supplementary material, dataset 1.

### 4.1.2. Genomic and transcriptomic data

We used the sequence read archive (SRA), a public repository for biological sequence data run by the National Center for Biotechnology Information (NCBI), to record the total amount of genomic and transcriptomic sequence data for all non-human primates (herein called genomic data) (https://www.ncbi.nlm.nih.gov/sra). SRA was searched using the broad taxonomic terms while purposefully excluding any data on humans. The search terms were as follows:

(primate OR primates) AND (genomic or genome or transcriptome or transcriptomic) NOT (Homo sapiens)

The final list of all deposits was then reviewed to remove any species that were misclassified as primates and extinct primate taxa to arrive at a full list of genomic or transcriptomic data deposits for all non-human primates. Ambiguous deposits that were not species-specific (e.g. deposits that were listed as only 'Rhinopithecus') were removed from the dataset. Genomic data from hybridized species were also removed from the dataset. All deposits that were removed from the study are listed in electronic supplementary material, table S3. The total amount of genomic data for each species was gathered on 16 August 2018.

### 4.1.3. Non-medical publications

Data for this variable were collected using a similar methodology as in Wiens (2016) [34] described below. We searched for the number of publications using either the species scientific name or a common name for the non-human primate species. An example of the search criteria for each of these variables is shown below.

Research intensity: TS = (('Allocebus trichotis') OR ('Hairy-eared Dwarf Lemur'))

The total number of publications was recorded for all species within the full non-human primate species list. These data were collected in January and February 2018. We then subtracted the number of medically focused publications from the number of total publications to compute this variable.

### 4.1.4. Importance in medical research

As discussed previously, non-human primates are frequently used in the testing of vaccinations, the study of SIV progression to inform HIV studies and for other research projects that have medical relevance. Importance in medical research was gauged using the number of papers published within Web of Science for each species under the Web of Science Category: Medicine, Research and Experimental using either the scientific name or one of the common names for each species. An example of the search criteria is shown below:

WC = (Medicine, Research and Experimental) AND TS = (('*Trachypithecus selangorensis*') OR ('Selangor Silvery Langur'))

The total number of publications for each species identified using this search was recorded. These data were collected in January and February 2018.

### 4.1.5. Frequency in captivity

We hypothesized that the more abundant a species was in captivity, the more opportunity there may have been for the collection of high-quality samples for genomic data analysis. To test this hypothesis, we used the number of individuals within a species found in captivity. These data were obtained from the Species 360 ZIMS database, an online repository of species currently held in captivity within institutions partnered with Species 360 around the world (https://www.species360.org/). Access to the Species 360 database was granted by the Duke Lemur Center. Each species was searched within the ZIMS database and the total number of individuals in captivity was recorded.

### 4.1.6. Relatedness to humans

We were interested in testing whether genomic data for species more closely related to humans were generated at disproportionately higher rates. Thus, for each species, we recorded how many millions

of years since they last shared a common ancestor with humans using the estimates generated in Dos Reis *et al.* [19].

### 4.1.7. Geographical distribution

Species with more extensive geographical distributions may be more easily accessible and therefore more frequently studied by scientists and conservationists. Geographical distribution for each available primate species was obtained from spatial data provided by the IUCN Red List. The spatial data can be accessed via the IUCN Red List online portal (https://www.iucnredlist.org/). The data were imported into ArcGIS and projected onto the Cylindrical Equal Area (sphere), where we then calculated the area of their range in square kilometres. These data were obtained on 7 February 2018.

### 4.1.8. IUCN Red List status

We also sought to test whether there is a relationship between an organism's perceived risk of extinction and the level of genomic sequence data generation. Red List status was obtained through the IUCN Red List of Threatened Species online portal (https://www.iucnredlist.org/). If Red List status could not be obtained via the online portal, the status was recorded from *All the World's Primates* by Rowe and Myers [25]. The possible categories for the Red List are as follows: data deficient (DD), least concern (LC), near threatened (NT), vulnerable (VU), endangered (EN) and critically endangered (CR). These data were collected in February 2018.

### 4.1.9. Activity pattern

Species that are diurnal or cathemeral (flexible day/night activity) may be easier to study than those that employ a nocturnal lifestyle. We hypothesized that species that were nocturnal would have less genomic data available, when controlling for the number of species within each of these categories. These data were collected using *All the World's Primates* by Rowe and Myers [25]. The four possible categories were: diurnal, nocturnal, cathemeral or N/A if no information was available.

### 4.1.10. Statistical analyses

Statistical analyses were conducted using RStudio. Code for all analyses performed is available via the GitHub repository https://github.com/maggiehern/PrimateGenomeProject and has been archived within the Zenodo repository: https://doi.org/10.5281/zenodo.4011305 [35]. The distribution of the amount of genomic data among non-human primates was right skewed and was normalized via log base 10 transformation prior to analyses. All continuous independent variables were also log transformed. Logistic regressions and generalized linear models were performed for the entire dataset and the subset of species with genomic data available. A list of models and their results are reported in electronic supplementary material, table S4.

## 4.2. Qualitative data and analyses

### 4.2.1. Data collection

We downloaded the metadata for all non-human primate deposits in the NCBI Sequence Read Archive (SRA). A list of SRA study numbers was generated by removing any SRA study numbers that had multiple entries. We randomly sampled 300 unique SRA study numbers identified in the linked publication using both the SRA study number and any other related accession number reported within SRA (this included the BioProject number, Gene Expression Omnibus deposition number, etc.) and confirmed that the paper represented the original generation of the data.

The human subjects research component of this study was approved by Penn State's Institutional Review Board (IRB) under the study number STUDY00008181. We contacted each of the corresponding authors of these studies to invite them to participate in a semi-structured interview. There were several papers that had the same corresponding author. In this case, a single email was sent listing all the papers we were requesting the authors to discuss during the interview. In total, we contacted 216 authors. There was one case where a corresponding author was not available for an interview and referred us to the first author of the paper, who consented to be interviewed. The authors were asked to participate in a 30 min interview following a semi-structured interview format using the questions listed

in electronic supplementary material, table S2. Interviews were conducted via Skype, Zoom, phone or in person at the participant's convenience. One survey was administered as a word document and filled out by the participant. Participants consented to being recorded prior to the start of the interview and were recorded using a Roland WAVE/MP3 Recorder R-05 and later transcribed.

### 4.2.2. Qualitative data analysis

We conducted a grounded theory analysis using all interviews conducted for this study [27]. Grounded theory analysis is an inductive approach to understanding qualitative data where researchers read over interview transcripts (or other texts) several times while developing successively more detailed levels of coding in order to understand major themes that emerge from the texts. In this case, we transcribed and then read over transcripts of the interviews to identify and code the major themes that emerged from the responses of the participants. Interviews were then reviewed again to record the frequency of each identified theme across all 33 interviews. This process was done in the software NVIVO, used for qualitative research data collection and organization. Major themes were then grouped into larger categories and subsequently compared to the factors identified in the quantitative portion of this project.

Ethics. The human subjects research component of this study was approved by Penn State's Institutional Review Board (IRB) under the study number STUDY00008181. As per our IRB agreement and to protect the confidentiality of our research participants, interviews from the qualitative portion of our work are not available. All metadata associated with our grounded theory analysis of the interview data are presented within the manuscript.

Data accessibility. Data and relevant code for this research work are stored in GitHub: https://github.com/maggiehern/PrimateGenomeProject and have been archived within the Zenodo repository: https://doi.org/10.5281/zenodo.4011305 [35]. As per our IRB agreement and to protect the confidentiality of our research participants, interviews from the qualitative portion of our work are not available. All metadata associated with our grounded theory analysis of the interview data are presented within the manuscript.

Authors' contributions. M.H. and G.H.P came up with the research project and design. M.H. carried out the research and analyses. G.H.P and M.K.S provided analytical advice. M.H. and G.H.P wrote the manuscript. M.H., G.H.P. and M.K.S edited the manuscript.

Competing interests. We declare we have no competing interests.

Acknowledgements. We thank the participants of this research study who took the time out of their days to speak with us and contribute to our project. Additionally, we thank Nicholas Triozzi, Dylan Davis, Kathleen Grogan, Christina Bergey, Julie White and Stephanie Marciniak for their discussion and analytical advice. We also thank Max Fancourt from the IUCN Red List and Sarah Zehr from the Duke Lemur Center for their assistance in data access. This material is based upon work supported by the National Science Foundation (NSF) Graduate Research Fellowship Program under grant no. DGE1255832 (to M.H.) and by NSF grant BCS-1554834 (to G.H.P.). Any opinions, findings and conclusions or recommendations expressed in this material are those of the authors and do not necessarily reflect the views of the National Science Foundation.

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
