## [Reviewer comments · Royal Society Open Science]

Review History

RSOS-201206.R0 (Original submission)

Review form: Reviewer 1

Is the manuscript scientifically sound in its present form?

Yes

Are the interpretations and conclusions justified by the results?

Yes

Is the language acceptable?

Yes

Do you have any ethical concerns with this paper?

No

Have you any concerns about statistical analyses in this paper?

Yes

Recommendation?

Accept with minor revision (please list in comments)

Comments to the Author(s)

The manuscript by Hernandez et al. provides a nice overview into distinct biases that exist in non-human primate genomics. The coupling of a quantitative approaches with interviews is interesting as it shows how global trends relate to the experiences of individual researchers. One ambivalent strength of the manuscript is that it repeatedly identifies many points, which would warrant further curiosity and a closer study - but does not provide more detail itself. I will describe some of these below but believe that addressing them fully might be outside of the scope of this publication. Methodologically, only one statistical mistake, which could be easily corrected, caught my eye. Stylistically, the manuscript could be cleaner.

Statistics:

The manuscript provides averages and a t-test to describe differences between those species with and without genomic information. As hinted at in their Supplemental Figures, the data itself however is not normal. Hence the non-parametric alternatives of median and two-sided Mann-Whitney U test would be appropriate.

Code:

The code neither tells how to import the data files (for which there would be different ways) nor does it contain lines of code to read the data files in (which would avoid ambiguity). This prevented me from testing.

Stylistic:

When referring to elements of a panel, it would be easier for the reader, if the order in the text matched the order of the panels.

A table with all individual considered predictors, and their associated R² and P-values (for both, presence of sequencing data, and extend of sequencing data for species with at least some data) would be helpful.

The choice of predictors is exiting as they are qualitatively diverse, and contain some that plausibly related directly to biological laboratory research and some that may provide justification of research that lies outside of the laboratory (e.g.: conservation). While hinted at, the manuscript does not state clearly what appears for me to be the main conclusion - namely that among the different plausible causes that could lead to genomic work on primates medical research is more informative than other plausible explanations. Possibly alternate hypothesis could be framed in the beginning of the manuscript.

The violin plots hide a lot of the data as most values are 0 and violin plots hold a kernel-estimate to smoothen the distribution. Would be useful to have the number of non-zero and zero-values indicated clearly (e.g.: stated in legend, or switching away from a violin plot to a non-parametric letter-value plot)

Additional curiosities:

While the manuscript follows an 1:M type inquiry between genomic information and other predictors, it would be interesting to see all (1+M):(1+M) comparisons between all predictors and genomic information. This would help to see if there are distinct groups of related features.

When considering multiple predictors simultaneously, the strength of the models is not visualized (e.g.: scatter plot between observed, and predicted), making it difficult to see, if there are systemic mis-predictions.

As there are likely non-linearities and possibly interactions or redundancies between predictors, an alternative analysis using Gradient Boosting Regression would appear to reveal a better quantification, and alternative way to assess the importance of individual predictors.

It is unclear, whether medical literature performs less well than the entire literature because it is medical, or whether there is fewer literature, which is medical – and hence numbers are smaller and more difficult to fit. A possible control analysis to distinguish would be to subsample papers from the entire literature randomly.

Do the findings hold for distinct modalities of omics data? E.g.: presence of a reference build of the genome, exome-sequencing, RNA-sequencing, ...

Which of the different modalities of omics dominates the data considered in this study?

Would the metadata of the experiments indicate whether the participants in the semi-structured interview used samples stemming from tissues of animals or from cell lines which could be cultured in a lab? How would the types of provided reasons change?

Since this is about genomics: How informative would it be whether there is an official accepted genome sequence of these primates? How informative would the years of the initial genome sequence builds be?

Review form: Reviewer 2

Is the manuscript scientifically sound in its present form?

Yes

Are the interpretations and conclusions justified by the results?

Yes

Is the language acceptable?

Yes

Do you have any ethical concerns with this paper?

No

Have you any concerns about statistical analyses in this paper?

No

Recommendation?

Accept as is

Comments to the Author(s)

I find paper "Factors influencing taxonomic unevenness in scientific research: A mixed-methods case study of non-human primate genomic sequence data generation" to be clearly written and the analyses done to be rational and competently performed. While I feel the results presented represent a confirmation of something almost anyone in the field would expect to be the case, it is nice to see it confirmed. Additionally, the grounded theory approach is an interesting way to explore the qualitative factors behind the taxonomic unevenness in scientific research.

Decision letter (RSOS-201206.R0)

Dear Mrs Hernandez

On behalf of the Editors, we are pleased to inform you that your Manuscript RSOS-201206 "Factors influencing taxonomic unevenness in scientific research: A mixed-methods case study of non-human primate genomic sequence data generation" has been accepted for publication in Royal Society Open Science subject to minor revision in accordance with the referees' reports. Please find the referees' comments along with any feedback from the Editors below my signature.

Both reviewers were very positive about publication. Reviewer 1 raised a number of interesting comments to improve the manuscript further and we invite you to respond to the comments and revise your manuscript. Below the referees' and Editors' comments (where applicable) we provide additional requirements. Final acceptance of your manuscript is dependent on these requirements being met. We provide guidance below to help you prepare your revision.

Please submit your revised manuscript and required files (see below) no later than 7 days from today's (ie 10-Aug-2020) date. Note: the ScholarOne system will 'lock' if submission of the revision is attempted 7 or more days after the deadline. If you do not think you will be able to meet this deadline please contact the editorial office immediately.

on behalf of Professor Steve Brown (Subject Editor)
openscience@royalsociety.org

Reviewer comments to Author:
Reviewer: 1

Comments to the Author(s)

The manuscript by Hernandez et al. provides a nice overview into distinct biases that exist in non-human primate genomics. The coupling of a quantitative approaches with interviews is interesting as it shows how global trends relate to the experiences of individual researchers. One ambivalent strength of the manuscript is that it repeatedly identifies many points, which would warrant further curiosity and a closer study - but does not provide more detail itself. I will describe some of these below but believe that addressing them fully might be outside of the scope of this publication. Methodologically, only one statistical mistake, which could be easily corrected, caught my eye. Stylistically, the manuscript could be cleaner.

Statistics:

The manuscript provides averages and a t-test to describe differences between those species with and without genomic information. As hinted at in their Supplemental Figures, the data itself however is not normal. Hence the non-parametric alternatives of median and two-sided Mann-Whitney U test would be appropriate.

Code:

The code neither tells how to import the data files (for which there would be different ways) nor does it contain lines of code to read the data files in (which would avoid ambiguity). This prevented me from testing.

Stylistic:

When referring to elements of a panel, it would be easier for the reader, if the order in the text matched the order of the panels.

A table with all individual considered predictors, and their associated R² and P-values (for both, presence of sequencing data, and extend of sequencing data for species with at least some data) would be helpful.

The choice of predictors is exiting as they are qualitatively diverse, and contain some that plausibly related directly to biological laboratory research and some that may provide justification of research that lies outside of the laboratory (e.g.: conservation). While hinted at, the manuscript does not state clearly what appears for me to be the main conclusion – namely that among the different plausible causes that could lead to genomic work on primates medical research is more informative than other plausible explanations. Possibly alternate hypothesis could be framed in the beginning of the manuscript.

The violin plots hide a lot of the data as most values are 0 and violin plots hold a kernel-estimate to smoothen the distribution. Would be useful to have the number of non-zero and zero-values indicated clearly (e.g.: stated in legend, or switching away from a violin plot to a non-parametric letter-value plot)

Additional curiosities:

While the manuscript follows an 1:M type inquiry between genomic information and other predictors, it would be interesting to see all (1+M):(1+M) comparisons between all predictors and genomic information. This would help to see if there are distinct groups of related features.

When considering multiple predictors simultaneously, the strength of the models is not visualized (e.g.: scatter plot between observed, and predicted), making it difficult to see, if there are systemic mis-predictions.

As there are likely non-linearities and possibly interactions or redundancies between predictors, an alternative analysis using Gradient Boosting Regression would appear to reveal a better quantification, and alternative way to assess the importance of individual predictors.

It is unclear, whether medical literature performs less well than the entire literature because it is medical, or whether there is fewer literature, which is medical – and hence numbers are smaller and more difficult to fit. A possible control analysis to distinguish would be to subsample papers from the entire literature randomly.

Do the findings hold for distinct modalities of omics data? E.g.: presence of a reference build of the genome, exome-sequencing, RNA-sequencing, ...

Which of the different modalities of omics dominates the data considered in this study?

Would the metadata of the experiments indicate whether the participants in the semi-structured interview used samples stemming from tissues of animals or from cell lines which could be cultured in a lab? How would the types of provided reasons change?

Since this is about genomics: How informative would it be whether there is an official accepted genome sequence of these primates? How informative would the years of the initial genome sequence builds be?

Reviewer: 2

Comments to the Author(s)

I find paper "Factors influencing taxonomic unevenness in scientific research: A mixed-methods case study of non-human primate genomic sequence data generation" to be clearly written and the analyses done to be rational and competently performed. While I feel the results presented represent a confirmation of something almost anyone in the field would expect to be the case, it is nice to see it confirmed. Additionally, the grounded theory approach is an interesting way to explore the qualitative factors behind the taxonomic unevenness in scientific research.

===PREPARING YOUR MANUSCRIPT===

- one version identifying all the changes that have been made (for instance, in coloured highlight, in bold text, or tracked changes);
- a 'clean' version of the new manuscript that incorporates the changes made, but does not highlight them.

 This version will be used for typesetting.

===PREPARING YOUR REVISION IN SCHOLARONE===

To revise your manuscript, log into <https://mc.manuscriptcentral.com/rsos> and enter your Author Centre - this may be accessed by clicking on "Author" in the dark toolbar at the top of the

page (just below the journal name). You will find your manuscript listed under "Manuscripts with Decisions". Under "Actions", click on "Create a Revision".

<https://royalsociety.org/journals/authors/author-guidelines/#supplementary-material> to include a suitable title and informative caption. An example of appropriate titling and captioning may be found at https://figshare.com/articles/Table_S2_from_Is_there_a_trade-off_between_peak_performance_and_performance_breadth_across_temperatures_for_aerobic_sc_ope_in_teleost_fishes_/3843624.

Author's Response to Decision Letter for (RSOS-201206.R0)

See Appendix A.

Decision letter (RSOS-201206.R1)

Dear Dr Hernandez,

It is a pleasure to accept your manuscript entitled "Factors influencing taxonomic unevenness in scientific research: A mixed-methods case study of non-human primate genomic sequence data generation" in its current form for publication in Royal Society Open Science.

on behalf of Professor Steve Brown (Subject Editor)
openscience@royalsociety.org

Appendix A

Response to reviewer comments:

We thank both reviewers for their thoughtful feedback on our manuscript. Below we describe how we incorporated the revisions into our manuscript and, if not, why we chose not to do so. Within this document, you will find our responses in red. We have also included the changes to the manuscript in red.

Reviewer 1:

Comments to the Author(s):

The manuscript by Hernandez et al. provides a nice overview into distinct biases that exist in non-human primate genomics. The coupling of a quantitative approaches with interviews is interesting as it shows how global trends relate to the experiences of individual researchers. One ambivalent strength of the manuscript is that it repeatedly identifies many points, which would warrant further curiosity and a closer study - but does not provide more detail itself. I will describe some of these below but believe that addressing them fully might be outside of the scope of this publication. Methodologically, only one statistical mistake, which could be easily corrected, caught my eye. Stylistically, the manuscript could be cleaner.

General Response to Reviewer 1: Thank you so much for your thoughtful feedback on both the quantitative and qualitative portions of the paper. Your feedback has made our manuscript stronger and clearer and we sincerely appreciate the time and effort you spent on reviewing our manuscript.

1. Statistics: The manuscript provides averages and a t-test to describe differences between those species with and without genomic information. As hinted at in their Supplemental Figures, the data itself however is not normal. Hence the non-parametric alternatives of median and two-sided Mann-Whitney U test would be appropriate.

Response: Thank you for this recommendation. We have accordingly replaced the t-tests with Mann-Whitney U tests, which we agree are more appropriate in this case. We observed a change in the result for one variable, millions of years since last shared common ancestor with humans, which went from being non-significant to significant using standard probability cutoffs. Please see the revised text below. In contrast, we decided to continue listing mean rather than median values in the manuscript text, due to informativeness. For example, the median values for number of medical publications for species with and without genomic data were zero.

“We found that species with genomic data, as a group, have significantly more non-medical publications (average $763.26 \pm 2,915.19$ s.d.) than those without genomic data (28.82 ± 74.42 ; Mann-Whitney U test; $P < 2.2 \times 10^{-16}$) (Supplementary Figure 3A). We also observed a significant difference in the number of medical publications between species with genomic data (16.17 ± 84.56) compared to those without genomic data (0.11 ± 1.08 ; $P < 2.2 \times 10^{-16}$) (Supplementary Figure 3B). Species with genomic data tended to be more closely related to humans (millions of years since last shared common ancestor with human; 45.03 ± 20.97) than those without genomic data (48.80 ± 17.17 ; $P = 0.0125$) (Supplementary Figure 3C). Finally species with genomic data available also had larger geographic ranges ($905,615 \pm 1,664,226$ km²) and more individuals in captivity (242 ± 502 individuals) than species without genomic data ($385,345 \pm 815,673$ km²; $P = 0.00429$; 28 ± 98 individuals; $P < 2.2 \times 10^{-16}$) (Supplementary Figure 3D and 3E).”

2. Code: The code neither tells how to import the data files (for which there would be different ways) nor does it contain lines of code to read the data files in (which would avoid ambiguity). This prevented me from testing.

Response: Thank you for pointing this out. We have added the code to be able to read in the data files.

3. Stylistic: When referring to elements of a panel, it would be easier for the reader, if the order in the text matched the order of the panels.

Response: We have revised the text to follow the order of the panels. See revised text in the answer above (this is the same text that has now been revised).

4. A table with all individual considered predictors, and their associated R2 and P-values (for both, presence of sequencing data, and extend of sequencing data for species with at least some data) would be helpful.

Response: Thank you for this suggestion. We have now added the results of the linear regressions within Supplementary Table 4, which also includes the statistics for the generalized linear models we performed within the paper. Below is the revised table.

Data	Test	Distribution	Variable(s) ¹	AIC	p-value	r ²
Whole dataset	Linear regression	Gaussian	Non-medical publications	N/A	<2.2x10 ⁻¹⁶	0.33
Species with genomic data	Linear regression	Gaussian	Non-medical publications	N/A	6.44x10 ⁻¹²	0.37
Whole dataset	Linear regression	Gaussian	Medical Publications	N/A	<2.2x10 ⁻¹⁶	0.27
Species with genomic data	Linear regression	Gaussian	Medical Publications	N/A	9.27x10 ⁻⁹	0.27
Whole dataset	Linear regression	Gaussian	Frequency in zoos	N/A	<2.2x10 ⁻¹⁶	0.22
Species with genomic data	Linear regression	Gaussian	Frequency in zoos	N/A	0.000222	0.12
Whole dataset	Linear regression	Gaussian	Relatedness to humans	N/A	5x10 ⁻⁶	0.038
Species with genomic data	Linear regression	Gaussian	Relatedness to humans	N/A	0.00106	0.092
Whole dataset	Linear regression	Gaussian	Geographical range	N/A	0.00032	0.028
Species with genomic data	Linear regression	Gaussian	Geographical range	N/A	0.0012	0.094
Whole dataset	Linear regression	Gaussian	IUCN Red List status	N/A	0.926	0.0021
Species with genomic data	Linear regression	Gaussian	IUCN Red List status	N/A	0.361	0.0016
Whole dataset	Logistic regression	Binomial	Non-medical publications + Relatedness to humans + Medical papers published + Geographical range + Frequency in zoos + IUCN Red List Status + Activity pattern	346.3	N/A	N/A
Whole dataset	GLM	Gaussian	Non-medical publications + Relatedness to humans + Medical papers published + Geographical range + Frequency in zoos + IUCN Red List Status + Activity pattern	1659.3	<2.2x10 ⁻¹⁶	0.40
Species with genomic data	GLM	Gaussian	Non-medical publications + Relatedness to humans + Medical papers published + Geographical range + Frequency in zoos + IUCN Red List Status + Activity pattern	293.26	6.16x10 ⁻⁸	0.40

Note:
¹ "+" indicates the function used for the variables within each model.

Supplementary Table 4. Analytical models performed. A list of all models used within the study, including data used, test performed, distribution of data, variables included, AIC values, p-values, and r² values where applicable.

5. The choice of predictors are as exciting as they are qualitatively diverse, and contain some that are plausibly related directly to biological laboratory research and some that may provide justification of research that lies outside of the laboratory (e.g.: conservation). While hinted at, the manuscript does not state clearly what appears for me to be the main conclusion – namely that among the different plausible causes that could lead to genomic work on primates medical research is more informative than other plausible explanations. Possibly alternate hypothesis could be framed in the beginning of the manuscript.

Response: Thank you for initiating this discussion and making this suggestion. While we didn't want to overemphasize the importance of medical research within the introduction, we definitely see your point on clarifying our hypotheses. We have revised the introduction text accordingly:

“Our goal was to identify variables associated with patterns of species-unevenness in genomic sequence data across all 519 non-human primate species. Are individual predictors (or combinations thereof) such as non-medical publication history, medical publication history, geographic range, frequency in captivity, International Union for the Conservation of Nature (IUCN) Red List conservation status, activity pattern, and phylogenetic distance to humans significantly associated with patterns of genomic data availability?”

6. The violin plots hide a lot of the data as most values are 0 and violin plots hold a kernel-estimate to smoothen the distribution. Would be useful to have the number of non-zero and zero-values indicated clearly (e.g.: stated in legend, or switching away from a violin plot to a non-parametric letter-value plot)

Response: This is a great point. Accordingly, we have now added the number of species that have no genomic data for each category of activity pattern in the legend of Supplementary Figure 2. Please see the revised text below.

“Supplementary Figure 2. Violin plots of per-species genomic data by activity pattern. Violin plot width corresponds to the density of species, which is also depicted via heatmap. There are 7 species that are cathemeral with no genomic data available, 290 species that are diurnal with no genomic data available, and 109 species that are nocturnal with no genomic data available. There are 11 species that are not represented because they did not have activity pattern information available.”

Additional curiosities:

7. While the manuscript follows an 1:M type inquiry between genomic information and other predictors, it would be interesting to see all (1+M):(1+M) comparisons between all predictors and genomic information. This would help to see if there are distinct groups of related features.

Response: Thank you for this suggestion. We have now conducted these analyses and we provide the results in a new figure, Supplementary Figure 3. We also included a line in the manuscript to direct our readers to these analyses. Below is the figure and the new text.

“These variables were compared to the amount of genomic data (Mb) available in the SRA database for each species, both on an individual variable basis (e.g. linear regressions) and collectively (e.g. logistic regression). We also performed linear regressions among all combinations of predictor variables (Supplementary Figure 3).”

Supplementary Figure 3. Linear regressions for all predictor variables. Plots showing the relationships between the different predictor variables used within our study. We show the linear regressions, including P and r^2 values, for each comparison in green where relevant. All plots, except for J, K, S, and T, show the density of species that occupy the space on the graphs. Darker red indicates more species, while light gray indicates less species. Plots J, K, S, and T show this through the jitter of points on the graph.

8. When considering multiple predictors simultaneously, the strength of the models is not visualized (e.g.: scatter plot between observed, and predicted), making it difficult to see, if there are systemic mis-predictions.

Response: This is an excellent idea. To address this point, we performed a leave-one-out cross validation test to determine the strength of the model. We have created a new figure with the observed and predicted values for the GLM of all predictor variables with the entire dataset as well as on the subset of data with species that have genomic data present. We have included these results within the manuscript in new text, shown below. We've also included a new figure, Supplementary Figure 4, where we show these results.

“To help assess our model, we performed the leave-one-out cross validation (LOOCV) analysis to compare predicted megabases of genomic data from our generalized linear model to the observed results (Supplementary Figure 4A). Predicted and observed values were significantly correlated (linear regression; $r^2 = 0.37$; $P < 2 \times 10^{-16}$). We obtained similar results when restricting the analysis to the subset of species with genomic data ($r^2 = 0.32$; $P = 5.93 \times 10^{-10}$; Supplementary Figure 4B).”

Supplementary Figure 4. Leave-one-out cross validation test. In order to test the strength of our GLMs, we performed a leave-one-out cross validation test for the entire dataset (A) and the subset of species that have genomic data present (B). To do this test, we iteratively left one sample out of the dataset and ran the GLM model. We then used the model to predict the megabases of genomic for the species that was left out. We ran models omitting each species in the dataset and plotted the observed values of megabases of genomic data (from our dataset) versus the predicted values (from the model) for each species. We then performed linear regressions on the observed versus the predicted values to see the strength of the models, reported in green.

9. As there are likely non-linearities and possibly interactions or redundancies between predictors, an alternative analysis using Gradient Boosting Regression would appear to reveal a better quantification, and alternative way to assess the importance of individual predictors.

Response: Thank you for this discussion. We did consider this point when originally designing our analyses, and we addressed it by removing redundancies in the predictor variables by making a distinction

between medical literature and total literature minus medical literature. We believe this approach addresses the concern. Please see the text from the manuscript below.

“The total number of publications was recorded for all species within the full non-human primate species list. These data were collected in January and February of 2018. We then subtracted the number of medically-focused publications from the number of total publications to compute this variable.”

10. It is unclear whether medical literature performs less well than the entire literature because it is medical, or whether there is fewer literature, which is medical – and hence numbers are smaller and more difficult to fit. A possible control analysis to distinguish would be to subsample papers from the entire literature randomly.

Response: Thank you for this recommendation. Unfortunately, due to the way the data were collected, we are not able to do a subsample on the number non-medical publications in order to directly compare our results to the medical publication analyses. However, to address this point we have revised the manuscript to mention this caveat:

“We note that there were many more non-medical than medical publications in our quantitative analysis. Thus, conclusions regarding the relative importance of these two variables should be considered tentative, given power differences.”

11. Do the findings hold for distinct modalities of omics data? E.g.: presence of a reference build of the genome, exome-sequencing, RNA-sequencing, ... Which of the different modalities of omics dominates the data considered in this study?

Response: This is an excellent question. For now, in part because we feel that this would then best be paired with expanded ethnographic work, we have chosen to keep all forms of the endogenous omics data together in our analyses. However, this topic is something that Anderson, Vilgays, and Tung 2020 touch on in their recent publication (<https://doi.org/10.1016/j.gde.2020.05.009>). We’ve incorporated their study into the discussion portion of our paper below.

“Future studies analyzing different sub-types of genomic data (e.g., shotgun genome sequencing, reduced representation sequencing, RNA-sequencing) may provide further insights into the potential drivers of taxonomic unevenness (e.g. 29).”

12. Would the metadata of the experiments indicate whether the participants in the semi-structured interview used samples stemming from tissues of animals or from cell lines which could be cultured in a lab? How would the types of provided reasons change?

Response: This is a great point, and something that was brought up regularly by interview participants. Interview participants did discuss how they obtained the samples that they used in order to sequence the genomic data for their species of choice. Their answers were given in the context of how easy it was to acquire these samples. For samples that were obtained from cell lines, for example, the scientists also tended to mention that it was much easier to acquire these than obtaining samples from the wild. This information was collapsed under the “ACCESS” theme. We added this point in the section where we

discuss the access theme in order to make your comment more visible within the manuscript. Please see revised text below.

“Opportunistic research: Authors frequently mentioned selecting species with sampling and analytical feasibility in mind. Four themes were categorized under opportunistic research: ACCESS (present in n=26 of 33 total interviews), HISTORY (n=23), CAPTIVE (n=10), and REFER (n=11). Having access to high-quality existing samples, the availability of easily-acquired cell lines, and/or access to captive individuals were repeatedly mentioned as being important.”

13. Since this is about genomics: How informative would it be whether there is an official accepted genome sequence of these primates? How informative would the years of the initial genome sequence builds be?

Response: We were also wondering the same thing while we were doing the research for this paper. Specifically, we aimed to test whether the presence of a reference genome influenced a change in the relative number of publications in the years preceding and following reference genome availability, or whether the pattern of the relative number of publications could be used to predict when a species would eventually have its genome sequenced. Unfortunately, there were too few species with reference genomes available in order to make a meaningful conclusion from the analysis we conducted. We hope that once there are more reference genomes available for non-human primate species, we may be able to explore this topic further. Thank you for the suggestion!

Reviewer 2:

Comments to the Author(s):

I find paper "Factors influencing taxonomic unevenness in scientific research: A mixed-methods case study of non-human primate genomic sequence data generation" to be clearly written and the analyses done to be rational and competently performed. While I feel the results presented represent a confirmation of something almost anyone in the field would expect to be the case, it is nice to see it confirmed. Additionally, the grounded theory approach is an interesting way to explore the qualitative factors behind the taxonomic unevenness in scientific research.

General Response to Reviewer 2: Thank you for your supportive comments. We, too, enjoyed incorporating the grounded theory approach and the mixed-methods aspect of this project!